# Optimal Dietary Intake of Riboflavin Associated with Lower Risk of Cervical Cancer in Korea: Korean National Health and Nutrition Examination Survey 2010–2021

**DOI:** 10.3390/life14040529

**Published:** 2024-04-20

**Authors:** Seon-Mi Lee, Aeran Seol, Hyun-Woong Cho, Kyung-Jin Min, Sanghoon Lee, Jin-Hwa Hong, Jae-Yun Song, Jae-Kwan Lee, Nak-Woo Lee

**Affiliations:** 1Department of Obstetrics and Gynecology, Korea University College of Medicine, 73 Koreadae-ro, Seongbuk-gu, Seoul 02841, Republic of Korea; tjsal4142@kumc.or.kr (S.-M.L.);; 2Department of Obstetrics and Gynecology, Korea University College of Medicine, 148 Gurodong-ro, Guro-gu, Seoul 08308, Republic of Korea; 3Department of Obstetrics and Gynecology, Korea University College of Medicine, 123 Jeokgeum-ro, Danwon-gu, Ansan-si 15355, Gyeonggi-do, Republic of Korea

**Keywords:** riboflavin, vitamin B2, vitamin B, vitamin, cervical cancer, cancer, Korean women

## Abstract

Background: This study aimed to evaluate the association between the dietary intake of vitamin B complex (thiamine, riboflavin, and niacin) and cervical cancer in Korea. Methods: The data from the Korean National Health and Nutrition Examination Survey (KNHANES) from 2010 to 2021 were analyzed, which included 28,306 participants who were categorized into non-cervical cancer and cervical cancer groups. The following dietary intake threshold levels of thiamine, riboflavin, and niacin were identified based on the recommended daily allowances (RDAs): thiamine, 1.1 mg/day; riboflavin, 1.2 mg/day; and niacin, 14 mg/day. Results: Among 28,306 participants, 27,976 were in the non-cervical cancer group and 330 were in the cervical cancer group. Riboflavin intakes of more than 1.2 mg/day but less than 2.4 mg/day were associated with a significantly reduced risk of cervical cancer, whereas intakes of above 2.4 mg/day were not associated with cervical cancer. Thiamine and niacin intakes were not significantly related to the risk of cervical cancer. Conclusions: The results of this study suggest that an intake of riboflavin of 1.2–2.4 mg/day may contribute to a lower risk of cervical cancer.

## 1. Introduction

Cervical cancer is the fourth most common female cancer, and according to 2020 statistics reported by the World Health Organization (WHO), there were approximately 604,000 new cases of cervical cancer diagnosed worldwide in 2020, and approximately 342,000 deaths from cervical cancer [1]. The Centers for Disease Control and Prevention (CDC) in the United States of America (USA) reported the following cervical cancer incidence trends across ethnicities worldwide. Moreover, the age-adjusted incidence of cervical cancer in 2019 was 7.7 per 100,000 women and the age-adjusted incidence of cervical cancer in 2020 was 6.8 per 100,000 women, thus indicating a slight decrease in the incidence of cervical cancer [2]. However, the age-adjusted mortality owing to cervical cancer in 2019 and 2020 remained similar at 2.2 per 100,000 women [2]. According to the analysis of the Korea National Cancer incidence database, the age-standardized incidence of gynecological cancers in 2019 was 8.0 per 100,000 for cervical cancer, 7.9 per 100,000 for uterine cancer, and 7.0 per 100,000 for ovarian cancer [3]. Among gynecological cancers, the incidence of cervical cancer was not only higher than those of other cancers, but it also was higher than the 2019 age-adjusted incidence of cervical cancer (7.7 per 100,000) for all races in the world [2,3]. Additionally, the age-standardized mortality rates for gynecological cancers in South Korea in 2019 were 1.7 per 100,000 for cervical cancer, 0.7 per 100,000 for uterine cancer, and 2.4 per 100,000 for ovarian cancer [3]. Compared to the age-adjusted cervical cancer mortality of 2.2 per 100,000 for the global population, the mortality rate for cervical cancer in South Korea was slightly lower [2]. As such, the incidence of cervical cancer in Korea is higher than that of the global population, and the mortality of cervical cancer is similar between the Korean and global populations. Therefore, we need to find ways to lower the risk of cervical cancer.

Recently, the role of B-complex vitamins in cellular life as essential components of DNA synthesis, DNA repair, and carbon metabolism as well as their ability to exert antioxidant effects by scavenging free radicals has become a focus of attention [4]. Among the B-complex vitamins, thiamin (vitamin B1), riboflavin (vitamin B2), and niacin (vitamin B3), which are not only essential cofactors in folic acid metabolism but also function as important intermediates in all cellular life [5], have attracted attention and several studies have been published on their effects on various cancers. In addition, the Korean National Health and Nutrition Examination Survey (KNHANES) data we used from 2010 to 2021 provided information on the participants’ daily intake of vitamins B1, B2, and B3, so we focused on vitamins B1, B2, and B3 in the vitamin B complex. The active form of thiamine (vitamin B1) is thiamine pyrophosphate (TPP), which serves as an important cofactor in the tricarboxylic acid cycle as part of energy production as well as in the mitochondria, where energy is metabolized via the conversion of fatty acids to acetyl coenzyme A (CoA) [5]. Studies have evaluated the relationship between thiamine and cancer, and some have reported its protective effects [6,7,8,9,10,11], whereas others have found no association between thiamine and cancer [12,13]. Riboflavin, in the form of flavin mononucleotide (FMN) and flavin adenine dinucleotide, contributes to redox reactions in the body and is essentially used by glutathione to destroy free radicals and perform liver detoxification [5]. In response to these antioxidant effects of riboflavin, several papers have suggested that riboflavin may help prevent several cancers [14,15,16,17,18,19]. However, some studies have found that riboflavin may contribute to the worsening of cancer [20,21], whereas others have found no association between riboflavin and cancer [12,22,23]. Niacin is a precursor of nicotinamide adenine dinucleotide (NAD) and nicotinamide adenine dinucleotide phosphate. NAD is necessary for maintaining genomic stability and performing base excision repair during DNA repair, and high NAD levels prevent the production of reactive oxygen species (ROS) [5]. Researchers have reported that niacin with these functions contributes to a meaningful reduction in the risk of cancer and cancer mortality [24,25,26], whereas others have reported that niacin is not associated with skin cancers such as basal cell carcinoma and melanoma [25].

The findings of these studies have been inconsistent. Although studies have evaluated the association between cervical cancer and B-complex vitamins, most were conducted in Western populations and evaluated the association between folate (vitamin B9), vitamin B12, or homocysteine (vitamin B6) and cancer. Additionally, few studies have evaluated the association between cervical cancer and thiamine (vitamin B1), riboflavin (vitamin B2), and niacin (vitamin B3) in Korean patients with cervical cancer. Therefore, the objective of this study was to evaluate the effect of thiamine, riboflavin, and niacin on the risk of cervical cancer in the Korean population.

## 2. Materials and Methods

### 2.1. Patient Selection and Data Collection

This study employed a cross-sectional design. All data are available from the KNHAES database (https://knhanes.kdca.go.kr/knhanes/sub03/sub03_02_05.do, accessed on 20 October 2023). In this research, the data were sourced from the KNHANES using the 2010–2021 datasets. KNHANES, initiated in 1998 and conducted every 3 years, aims to evaluate the health status, dietary habits, and nutritional conditions of the Korean population. The survey employs a multi-faceted approach, and includes the measurements of height, weight, waist circumference, and body mass index (BMI), along with extensive health-related questionnaires. These procedures are executed by medical staff who are extensively trained to ensure the accuracy and reliability of the data collected.

The inclusion criteria for this study were females aged over 20 years who completed the cervical cancer health questionnaire. The exclusion criteria were males, females aged <20 years, and incomplete data, such as incomplete questionnaires or missing values in the system. In total, 28,306 of 95,310 participants were selected for this study (Figure 1). The participants were divided into two groups: the non-cervical cancer and cervical cancer groups.

### 2.2. Study Variables

KNHANES includes demographic data, health-related factors, medical history, health examination results, education, economic activities, exanimation surveys, blood pressure, blood tests, body measurements, and nutrition surveys regarding dietary habits and nutritional information.

In this study, we categorized personal income into four distinct quartiles based on the average monthly household income adjusted for house size using the following formula: ([monthly overall household income] × [household size]^−0.5^). Based on income quartiles, in increasing order, the participants were categorized into low, lower-intermediate, upper-intermediate, and high-income groups. The educational level of participants was stratified into the following four groups: less than elementary school, middle school, high school, and college education or higher. In adult females, alcohol consumption patterns were examined and classified into the following five categories based on the frequency of drinking: not drinking at all, less than once a month, approximately once a month, approximately once a week, and almost every day. Based on a questionnaire regarding the lifetime smoking pattern of adults, participants were classified into non-smokers and smokers. The waist circumference was assessed by measuring the waist at the narrowest point between the rib cage and the iliac crest following normal expiration. This method ensures a standardized approach across participants. BMI was estimated using the following formula: weight in kg/height in m^2^.

We examined marital status and cohabitation. Participants who had a spouse but were not cohabiting with them, including those who were divorced or widowed, were categorized into the ‘not living together’ group. Conversely, those who were living with their spouse were classified into the ‘living together’ group. Regarding menstrual status, participants were grouped based on their responses into two categories: the ‘menstruation group’ for those who were still menstruating, and the ‘menopause group’ for those who had experienced natural or artificial menopause. Age at first childbirth and the number of pregnancies were also included in the questionnaire. In terms of contraceptive use, the participants were divided into ‘yes’ and ‘no’ groups based on their use of oral contraceptives.

In the KNHANES survey, a 24 h recall survey is conducted to collect information regarding food intake. The information regarding the food consumed is analyzed for each nutrient according to the 10th edition of the Food Composition List published by the Ministry of Agriculture and Rural Development. As a result, the types and amounts of nutrients consumed by an individual during the day can be identified. From 2010 to 2021, the questionnaire was used to determine the types and amounts of nutrients consumed by individuals over a 24 h period, with slight differences between each year. From 2010 to 2021, the survey content for the nutritional intake survey included all foods consumed on all days of the week as well as on weekends to analyze the types and amounts of nutrients. However, in 2021, owing to a change in the working conditions of the survey team, the food intake survey was conducted only on weekdays from Monday to Friday, and the contents of meals consumed on the weekends were not investigated; therefore, the food survey items from 2021 do not reflect the nutrient information for the weekends. In our study, we used the data from KNHANES from 2010 to 2021 to determine the amounts of vitamin B1, B2, and B3 in an individual’s 24 h intake. The recommended daily allowances (RDAs) for each of these B vitamins are as follows: thiamine (vitamin B1), 1.1 mg/day; riboflavin (vitamin B2), 1.2 mg/day; and niacin (vitamin B3), 14 mg/day. Considering these RDAs, the intake of thiamine was categorized into three categories: less than 1.1 mg/day, more than 1.1 mg/day but less than 2.2 mg/day, and more than 2.2 mg/day. The intake of riboflavin was categorized into three categories: less than 1.2 mg/day, more than 1.2 mg/day but less than 2.4 mg/day, and more than 2.4 mg/day. The intake of niacin was categorized into three categories: less than 14 mg/day, more than 14 mg/day but less than 28 mg/day, and more than 28 mg/day.

### 2.3. Statistical Analysis

This study utilized the KNHANES data, which were extracted via a two-stage stratified cluster sampling method. The statistical analysis was adapted to this complex sampling design. Continuous variables were compared using Student’s *t*-test, whereas categorical variables were analyzed using either the Chi-square or Fisher’s exact test. Univariate analysis was used to assess the differences in the risks between the cervical and non-cervical cancer groups. A multivariate logistic regression analysis, adjusted for age, age at first childbirth, and number of pregnancies, evaluated cervical cancer risk in relation to the consumption of these vitamin B types. All statistical analyses were performed using SPSS (v25.0; SPSS Inc., Chicago, IL, USA), with *p*-values < 0.05 considered significant.

### 2.4. Ethics

Access to the KNHANES data was granted following Institutional Review Board (IRB) approval from the Korea Center for CDC. This retrospective study utilized KNHANES survey data and, therefore, did not require additional IRB approval. An ethical exemption was warranted as the dataset was devoid of personal information, and the participants’ consent was obtained by KNHAENS, which aligns with IRB protocols for open data usage.

## 3. Results

The overall average age of participants included in this study was 51.74 ± 61.79 years (20–80 years old). Of the 28,306 participants, 27,976 and 330 were included in the non-cervical cancer and cervical cancer groups, respectively. A comparison of the characteristics between the two groups and the results of the univariate logistic regression analysis regarding the risks for cervical cancer are presented in Table 1. The mean age, BMI, waist circumference, and number of pregnancies of the participants were higher in the cervical cancer group than in the non-cervical cancer group, and the age at first childbirth was significantly lower in the cervical cancer group (23.91 ± 3.50 years) than that in the non-cervical cancer group (25.40 ± 4.21 years). In contrast, height was greater in the non-cervical cancer group than in the cervical cancer group. There were no significant differences in income, smoking, and alcohol consumption between the two groups. Daily energy intake was slightly higher in the non-cervical cancer group than in the cervical cancer group, but the result of univariate logistic regression was not statistically significant. In order to determine the activity level of the participants, this study used the variables of the number of days per week of walking and strength training, which are also shown in Table 1, and there was no significant difference between the two groups. On comparing the daily intake of B vitamins, the proportion of women who consumed less than 1.1, 1.2, and 14 mg/day of thiamine, riboflavin, and niacin, respectively, was higher in the cervical cancer group than that in the non-cervical cancer group. According to the univariate analysis, factors significantly associated with the risk of cervical cancer included older age, low height, high BMI, high waist circumference, and a high number of pregnancies. In contrast, a lower risk of cervical cancer was related to older age at first childbirth. Compared with consuming less than the RDAs of B vitamins, consuming more than 1.1 mg/day but less than 2.2 mg/day of thiamine, more than 1.2 mg/day but less than 2.4 mg/day of riboflavin, and more than 14 mg/day but less than 28 mg/day of niacin were associated with a lower risk of cervical cancer.

In the multivariate logistic regression analysis, after adjustments for age, BMI, age at first childbirth, number of pregnancies, energy intake, days of walking in a week, and days of strength training in a week, a riboflavin intake of more than 1.2 mg/day but less than 2.4 mg/day was inversely associated with the risk of cervical cancer (Table 2); however, riboflavin intake of more than 2.4 mg/day was not associated with the risk of cervical cancer. Additionally, neither thiamine intake above the RDA of 1.1 mg/day nor niacin intake above the RDA of 14 mg/day was associated with any risk of cervical cancer.

## 4. Discussion

In this retrospective study, we found that the daily intake of appropriate amounts of riboflavin (1.2 mg ≤ riboflavin intake < 2.4 mg/day) was associated with a significantly lower risk of cervical cancer. In contrast, other B vitamins, such as thiamine and niacin were not associated with any risk of cervical cancer. To the best of our knowledge, this study is one of the first to determine the association between the risk of cervical cancer and the daily intake of B vitamins (thiamine, riboflavin, and niacin) in Korean women.

Thiamine plays an important function as an intermediate in cellular energy production. It not only activates nucleic acid ribose synthesis through the non-oxidative transketolase-catalyzed pentose cycle reaction in cancer cells but also contributes to the proliferation of cancer cells by increasing energy production in the tumor cell environment [5,27]. Thiamine is mainly found in foods like bread, fish, pork, eggs, legumes, milk, yogurt, and cereals [28]. Westernized diets, especially those dominated by the consumption of flour-based foods, peanut butter, refreshment drinks, and pasta, provide an overabundance of thiamine [27,28]. Conversely, Asian and African diets, which are laden with raw fish, fermented fish, roasted insects, and certain clams containing thiaminase, have lower thiamine intakes [27,29]. This dietary variation might contribute to the higher cancer incidence in Western countries [27]. Studies in mice with Ehrlich’s ascites tumors show that consuming 12.5–205 times the recommended thiamine allowance led to a 164% increase in tumor growth, thus implicating thiamine in cancer cell growth [8]. However, thiamine, a cofactor in carbon metabolism, aids in purine synthesis for DNA and may prevent cancer by countering DNA methylation that leads to carcinogenesis [12]. A study in rats reported a significant increase in colonic aberrant crypt foci in the presence of thiamine deficiency [9], and another study in breast cancer cells reported that thiamine supplementation inhibited cancer cell metabolism and growth in breast cancer cell lines [11]. Additionally, a retrospective cohort and case–control study revealed that thiamine significantly reduced cancer growth in patients with HER2-positive breast cancer in Q3 (HR, 0.38; 95% CI, 0.54–0.92) and Q4 (HR, 0.31; 95% CI, 0.10–0.99) [6] and showed a strong negative relationship with prostate cancer in patients over 50 years of age [7]. Lee’s review also indicated that subcutaneous thiamine injections significantly reduced tumor growth in patients with severe malnutrition-related cancers, such as Baker’s cyst and osteosarcoma, leading to a cure without recurrence [10]. In contrast, one cohort study found no association between thiamine and the five major cancers in women: breast, endometrial, ovarian, colon, and lung cancers [12], the other cohort study, conducted in Canadians, also found no association between thiamine and breast, endometrial, ovarian, and colorectal cancers [13]. The results of these two studies are consistent with the results of our study. Unlike our study, which only included patients with cervical cancer, the previous studies were conducted on a variety of cancers such as breast, colon, prostate, ovarian, and so on. Although our study did not demonstrate any association between thiamine and cervical cancer, the function of thiamine in the cellular environments of various cancers may be expressed in different ways and further studies are needed to identify the precise molecular mechanisms.

Our results demonstrated that moderate riboflavin intake (1.2 to <2.4 mg/day) significantly reduces cervical cancer risk. Other studies have also indicated a decrease in colorectal cancer risk with riboflavin supplementation [14,15,16,17]. A Dutch study found an inverse relationship between riboflavin and colorectal adenoma [19], and lower plasma riboflavin was linked to worse outcomes in esophageal cancer [18]. Unlike our food source analysis, Lie et al. and Pan et al. used supplement-based and plasma riboflavin levels, respectively [15,18]. Similar to our method, Zschabitz et al., De Vogel et al., and other European authors used dietary riboflavin content to assess any cancer relationship [14,16,17,19]. Zschabitz et al. reported a colorectal cancer OR of 0.97 (95% CI, 0.50–1.17) for intakes of 1.80–2.87 mg/day, which was comparable to our OR of 0.76 (95% CI, 0.58–0.992) for intakes of 1.2–2.4 mg/day; however, Zschabitz et al. reported a lower OR of 0.81 (95% CI, 0.66–0.99) with higher intakes (>3.98 mg/day), which differed from our results [14]. These differences might be owing to varying riboflavin intake categories and reference standards. Although a clear mechanism for the interaction between riboflavin and cancer has not been identified, a summary of possible responses to riboflavin deficiency in the body is presented in Figure 2, which may support a negative relationship between riboflavin and cancer development. Riboflavin deficiency can cause protein and DNA oxidation, leading to DNA damage, cell cycle arrest, increased oxidative stress, and abnormal immune and inflammatory responses, potentially promoting carcinogenesis [5,30]. Some studies have reported an association between riboflavin deficiency and hepatocellular cancer as well as liver hypertrophy [31] and esophageal cancer [32]. In a study by Aili et al. evaluating the association between riboflavin levels in cervical tissue and plasma and cervical intraepithelial neoplasia (CIN) and cervical cancer, riboflavin levels in tissue and plasma were decreased in patients with CIN and cervical cancer compared to patients with normal cervices. In particular, the level of riboflavin tended to decrease more in advanced-stage cervical cancer, and the level of riboflavin decreased even more in cervical tissue infected with human papillomavirus (HPV) 16 and 18. In addition, the expression of C20orf54 protein, which functions as a transporter of riboflavin, was significantly increased in CIN and cervical cancer compared to normal cervical tissue [33]. Similarly, Ma’s study showed that the methylation of the promoter gene of human riboflavin transporter (hRFT2), which acts as a transporter of riboflavin, was increased in cervical cancer tissue. The expressed hRFT2 protein was found to be translocated from the cytoplasm to the nucleus [34]. These intracellular mechanisms may provide an important key to understanding the effect of riboflavin on cervical carcinogenesis, which is still unclear. It is necessary to further investigate whether riboflavin transporters, such as C20orf54 or hRFT2, can play a protective effect in the environment of cervical cancer. In contrast, Ma et al. found that high riboflavin levels increased the risk of colon cancer [21], and Yang reported accelerated lung cancer cell growth with higher doses of riboflavin [20]. These findings suggest that high levels of riboflavin may promote cancer cell invasion by increasing extracellular matrix components and matrix metalloproteinases [20,21]. However, other studies have reported no significant association between riboflavin and colorectal cancer, various types of female cancer, and endometrial cancer [12,22,23]. Although we have not identified any studies showing that the association between riboflavin and cancer is driven by racial differences in outcomes, additional research is needed to determine whether the association may be driven by racial differences in outcomes.

No significant relationship was found between niacin and cervical cancer in this study. Similarly, a cohort study that evaluated the association between niacin intake in the form of food or supplements and skin cancer reported that the HRs adjusted for family history of melanoma, sun exposure, and exercise time did not reveal a statistically significant increase in the risk of skin cancer with increasing intake in Q2 (HR 1.02; 95% CI, 0.80–1.30), Q3 (HR, 1.02; 95% CI, 0.55–1.89), Q4 (HR, 0.91; 95% CI, 0.73–1.14), and Q5 (HR, 1.18; 95% CI, 0.77–1.81) compared to Q1 [25]. However, other studies have reported a negative association between niacin and the incidence of various cancers [24,26]. Ying et al. evaluated the association between dietary niacin intake and cancer mortality in patients diagnosed with various cancers in the National Examination Survey and observed a significant decrease in cancer mortality with increasing niacin intake in model 1, which was adjusted for age, sex, race, and BMI, as well as in model 2, which was adjusted for the aforementioned factors as well as energy intake, nutrient intake, and physical activity, all with niacin intake Q1 as a reference [24]. A case–control study in a Chinese population evaluated the association between the intake of B vitamins and glioma and found that the risk of developing glioma was reduced significantly with increasing niacin intake [26]. These studies suggest that niacin promotes DNA repair and may inhibit cancer growth by inhibiting pro-inflammatory mediators [24,26]. Particularly, several studies have reported that in the environment of glioma cells, niacin not only degrades F-actin stress fibers, causing cellular matrix adhesion but also promotes snail-1 degradation, causing intracellular adhesion, which can inhibit the invasion and metastasis of glioma cells [35,36]. However, unlike glioma, which occurs in the presence of multiple neural stem cells and oligodendrocyte precursor cells [37], the cervix is composed of nonkeratinizing squamous and columnar epithelium [38]; so, it cannot be concluded that the function of niacin expressed in glioma cells will be the same in the cervix. Therefore, the role of niacin in cervical cancer cells, whether it inhibits cancer growth or promotes cancer progression, needs to be further investigated.

This study has some limitations. First, this was a cross-sectional study, and we were unable to perform follow-up. Additionally, a cross-sectional study evaluates causes and results at a single time point and analyzes their association; therefore, it is difficult to establish a general causal relationship. Second, participants who did not complete the questionnaire or were missing data were excluded. Of 51,858 women in this study, we excluded 9798 women under the age of 20 years and 13,574 women with missing values, which is approximately 45% of all women in the KNHANES data during 2010–2021. Consequently, a large number of participants were excluded, which may have resulted in exclusion bias. Third, due to the characteristics of the KNHANES data, which are collected in the form of a survey, participants are required to rely on their memories to respond, which may have included a recall bias. Fourth, the KNHANES data provide information on whether cervical cancer was diagnosed, but it is not possible to determine the cellular type of cervical cancer (squamous cell carcinoma or adenocarcinoma). For this reason, the effect of B vitamins in different cancer cell environments could not be assessed in cervical cancer. Fifth, although HPV infection is a well-known and significant risk factor for cervical cancer, information regarding HPV infection was not available in the KNHANES data; therefore, we were unable to evaluate the association between B vitamins and cervical cancer based on HPV infection status. Sixth, as outlined in the Section 2, we could not gather weekend food intake data from 2021. This was owing to changes in the 24 h food intake survey from 2010 to 2020 and subsequent changes in 2021. This may have introduced sample bias. Nevertheless, the strength of our study is that it is one of the few studies to evaluate the association between cervical cancer and B vitamins, such as vitamins B1, B2, and B3, using a large nationally representative sample dataset of 12 years for Korean female cervical cancer patients. Consequently, we identified a significant negative association between moderate riboflavin intake and cervical cancer. Longitudinal follow-up studies should be conducted to evaluate the causal relationship between riboflavin intake and cervical cancer or whether riboflavin intake affects the outcome of cervical cancer.

## 5. Conclusions

In our study, among vitamins B1, B2, and B3, appropriate riboflavin intake of more than 1.2 mg/day but less than 2.4 mg/day significantly lowered the incidence of cervical cancer in Korean women. Therefore, encouraging optimal intake of riboflavin above the RDA in women may contribute to lowering the potential risk of cervical cancer in the future.

## Figures and Tables

**Figure 1 life-14-00529-f001:**
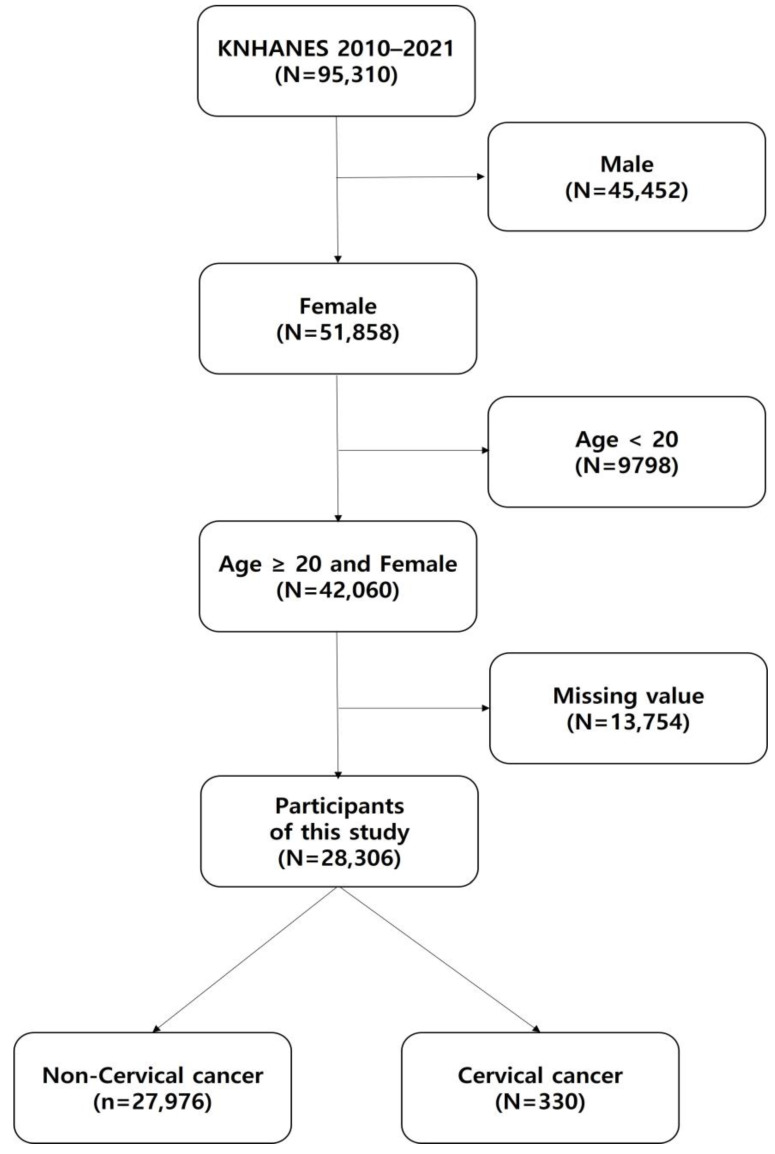
A diagram of participants for final analysis. KNHANES: Korea National Health and Nutrition Examination Survey.

**Figure 2 life-14-00529-f002:**
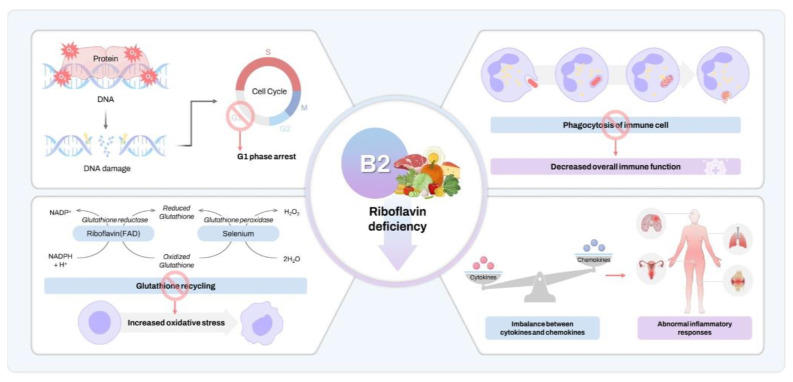
Schematic representation of the body’s response to riboflavin deficiency.

**Table 1 life-14-00529-t001:** Comparison of characteristics between the two groups and risk of cervical cancer compared to non-cervical cancer via univariate logistic regression analysis.

	Non-Cervical Cancer (N = 27,976)	Cervical Cancer (N = 330)	*p*-Value	Risk for Cervical Cancer Compared to Non-Cervical Cancer		
				OR	95% CI	*p*-Value
Age (year)	51.63 ± 16.80	60.80 ± 13.34	<0.001	1.04	(1.029, 1.043)	<0.001
Weight, kg	58.13 ± 9.63	58.30 ± 9.43	0.757	1.01	(0.991, 1.013)	0.757
Height, cm	157.25 ± 6.63	154.88 ± 6.03	<0.001	0.95	(0.934, 0.964)	<0.001
BMI, kg/m^2^	23.52 ± 3.65	24.25 ± 3.46	<0.001	1.05	(1.023, 1.080)	<0.001
Waist circumference, cm	79.63 ± 10.08	82.02 ± 10.03	<0.001	1.02	(1.012, 1.033)	<0.001
Income, N (%) Q1 Q2 Q3 Q4	6849 (24.5%) 7031 (25.1%) 7051 (25.2%) 7045 (25.2%)	97 (28.9%) 89 (25.8%) 82 (25.2%) 65 (20.1%)	0.108	1.00 0.87 0.85 0.67	Reference (0.647, 1.171) (0.629, 1.142) (0.489, 0.924)	0.359 0.277 0.014
Education, N (%) Less than Elementary school Middle school High school College or higher	7161 (25.6%) 2837 (10.1%) 8741 (31.2%) 9237 (33.1%)	154 (46.2%) 41 (12.5%) 91 (27.8%) 44 (13.5%)	<0.001	1.00 0.69 0.49 0.23	Reference (0.488, 0.978) (0.380, 0.641) (0.161, 0.316)	0.037 <0.001 <0.001
Alcohol consumption, N (%) Not drinking at all Less than once a month About once a month About once a week Almost every day	10,585 (48.7%) 6053 (22.8%) 4657 (14.9%) 4076 (11.0%) 2605 (2.6%)	15 (57.8%) 63 (19.1%) 47 (9.8%) 51 (12.1%) 12 (1.2%)	0.077	1.00 0.70 0.56 0.93 0.38	Reference (0.474, 1.046) (0.332, 0.933) (0.581, 1.496) (0.093, 1.541)	0.082 0.026 0.771 0.175
Smoking, N (%) Non-smoker Smoker	24,775 (88.8%) 3201 (11.2%)	292 (89.0%) 38 (11.0%)	0.922	1.00 0.98	Reference (0.694, 1.392)	0.922
Menstrual status, N (%) Menstruating Menopausal	14,922 (53.4%) 13,054 (46.6%)	148 (44.4%) 182 (55.6%)	0.002	1.00 1.44	Reference (1.144, 1.803)	0.002
Living status with spouse, N (%) Not living together Living together	7535 (23.1%) 20,441 (76.9%)	114 (34.2%) 216 (65.8%)	<0.001	1.00 0.58	Reference (0.485, 0.731)	<0.001
Oral contraceptive use, N (%) Yes No	4791 (16.9%) 23,185 (83.1%)	71 (21.2%) 259 (78.8%)	0.041	1.00 0.76	Reference (0.579, 0.990)	0.042
Age at first childbirth, years	25.10 ± 4.21	23.91 ± 3.50	<0.001	0.91	(0.882, 0.937)	<0.001
Number of pregnancies, N (%)	3.76 ± 2.06	4.38 ± 2.37	<0.001	1.12	(1.072, 1.168)	<0.001
Energy intake (Kcal)/day	1649.88 ± 690.14	1570.10 ± 677.56	0.046	0.99	(0.999, 1.009)	0.054
Days of walking in a week	5.35 ± 7.79	5.38 ± 9.35	0.938	1.001	(0.987, 1.014)	0.938
Days of strength training in a week	1.55 ± 1.37	1.54 ± 1.42	0.886	0.994	(0.918, 1.077)	0.886
Thiamine intake (mg)/day Intake < 1.1 mg 1.1 mg ≤ Intake < 2.2 mg Intake ≥ 2.2 mg/day	13,339 (49.4%) 10,888 (39.7%) 3749 (10.9%)	181 (56.6%) 112 (34.1%) 37 (9.3%)	0.046	1.00 0.75 0.74	Reference (0.587, 0.961) (0.497, 1.110)	0.023 0.147
Riboflavin intake (mg)/day Intake < 1.2 mg 1.2 mg ≤ Intake < 2.4 mg Intake ≥ 2.4 mg	14,128 (52.6%) 10,904 (39.7%) 2944 (7.7%)	205 (64.6%) 96 (28.8%) 29 (6.6%)	<0.001	1.00 0.59 0.70	Reference (0.459, 0.763) (0.443, 1.118)	<0.001 0.137
Niacin intake (mg)/day Intake < 14 mg 14 mg ≤ Intake < 28 mg Intake ≥ 28 mg	18,809 (71.5%) 7408 (25.6%) 1759 (2.9%)	244 (77.5%) 71 (20.5%) 15 (2.0%)	0.067	1.00 0.74 0.63	Reference (0.558, 0.979) (0.279, 1.422)	0.035 0.266

Note: Values are presented as means ± standard deviation or non-weighed N (weighted %). BMI, body mass index; N, number; OR, odds ratio; CI, confidence interval.

**Table 2 life-14-00529-t002:** Multivariate logistic regression analysis of cervical cancer risk by daily intake of thiamine (vitamin B1), riboflavin (vitamin B2), and niacin (vitamin B3).

Risk of Cervical Cancer Compared to Non-Cervical Cancer			
	OR	95% CI	*p*-Value
Thiamine intake (mg)/day Intake < 1.1 mg mg ≤ Intake < 2.2 mg Intake ≥ 2.2 mg	1.00 0.76 0.66	Reference (0.569, 1.003) (0.394, 1.113)	0.052 0.120
Riboflavin intake (mg)/day Intake < 1.2 mg mg ≤ Intake < 2.4 mg Intake ≥ 2.4 mg	1.00 0.74 0.89	Reference (0.546, 0.991) (0.494, 1.588)	0.043 0.684
Niacin intake (mg)/day Intake < 14 mg 14 mg ≤ Intake < 28 mg Intake ≥ 28 mg	1.00 0.89 0.94	Reference (0.634, 1.257) (0.385, 2.290)	0.516 0.890

Multivariate logistic regression analysis adjusted for age. OR, odds ratio; CI, confidence interval.

## Data Availability

Data and materials are available on reasonable request. The raw data of KNHANES used in this paper can be accessed through the following website: https://knhanes.kdca.go.kr/knhanes/sub03/sub03_02_05.do, accessed on 20 October 2023.

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
