# Peer review of "Optimal Dietary Intake of Riboflavin Associated with Lower Risk of Cervical Cancer in Korea: Korean National Health and Nutrition Examination Survey 2010–2021"

_life, 2024, doi:10.3390/life14040529_

Round 1

Reviewer 1 Report

Comments and Suggestions for Authors

Dear Authors,

I read your work and I really appreciate your study. However I have few comments that may improve your manuscript. 

TThe abstract is concise and well written.

The introduction is detailed and adequated to the subject.

The results are well exposed with figures and tables.

În discussion section I have one concert regarding The figure. Do you need any copyright? Is it self image? 

I recommended that the discusion to be made based on Prisma statement.

The conusions are adequated.

Author Response

I would like to submit my answer as a word file. 

Thank you. 

Sincerely, 

Reviewer 2 Report

Comments and Suggestions for Authors

It seems that a comprehensive study has not been done.

It is not clear why these three vitamins were chosen and the risk for intake of all micro-nutrients was not considered by adjusting for other micro-nutrients, especially energy.

Statistical analysis with adjustment for intervening variables, especially energy, body mass index and physical activity is recommended.

Despite the small number of tables, why is one table included in the appendices? It is better to move it to the main file.

Comments on the Quality of English Language

-

Author Response

(The authors gave the same response as above.)

Reviewer 3 Report

Comments and Suggestions for Authors

This article focuses on an important topic related to the clinical implications of riboflavin in cervical cancer pathogeny. Since the literature has scarce data, this study is essential in identifying the mechanisms that will underlie the best experimental and clinical practices.

However, some suggestions could improve the quality of the article:

- Line 185 to specify the age range

- If there are racial/ethnic differences regarding the association of riboflavin with cervical cancer?

- In the Discussions chapter, it would be recommended to mention the possibility of establishing the level of riboflavin in plasma (chromatographic profile) or the fresh tissue of patients with cervical cancer or riboflavin transporter 2 and the risk of cancer.

- Insufficient riboflavin was associated with an increased risk of cervical dysplasia and persistence of HPV infection.

Kind regards

Author Response

I would like to submit my answer as a word file. 

Thank you. 

Round 2

Reviewer 2 Report

Comments and Suggestions for Authors

Statistical analysis must be corrected with adjustment for intervening variables, especially energy, body mass index and physical activity.

It is better to mention the adjusted variables on the footnote of Table 2.

Comments on the Quality of English Language

-

Author Response

(The authors gave the same response as above.)
